# Within-Generation Polygenic Selection Shapes Fitness-Related Traits across Environments in Juvenile Sea Bream

**DOI:** 10.3390/genes11040398

**Published:** 2020-04-07

**Authors:** Carine Rey, Audrey Darnaude, Franck Ferraton, Bruno Guinand, François Bonhomme, Nicolas Bierne, Pierre-Alexandre Gagnaire

**Affiliations:** 1ISEM, Univ Montpellier, CNRS, EPHE, IRD, Montpellier, France; carine.rey@ens-lyon.fr (C.R.); bruno.guinand@umontpellier.fr (B.G.); francois.bonhomme@umontpellier.fr (F.B.); nicolas.bierne@umontpellier.fr (N.B.); 2Univ Lyon, ENS de Lyon, Univ Claude Bernard, CNRS UMR 5239, INSERM U1210, Laboratoire de Biologie et Modélisation de la Cellule, 15 parvis Descartes, F-69007 Lyon, France; 3MARBEC, Univ Montpellier, CNRS, IRD, Ifremer, 34095 Montpellier, France; audrey.darnaude@umontpellier.fr (A.D.); franck.ferraton@umontpellier.fr (F.F.)

**Keywords:** antagonistic pleiotropy, habitat association, fitness trade-off, juvenile growth, polygenic scores, RAD-sequencing, spatially varying selection

## Abstract

Understanding the genetic underpinnings of fitness trade-offs across spatially variable environments remains a major challenge in evolutionary biology. In Mediterranean gilthead sea bream, first-year juveniles use various marine and brackish lagoon nursery habitats characterized by a trade-off between food availability and environmental disturbance. Phenotypic differences among juveniles foraging in different habitats rapidly appear after larval settlement, but the relative role of local selection and plasticity in phenotypic variation remains unclear. Here, we combine phenotypic and genetic data to address this question. We first report correlations of opposite signs between growth and condition depending on juvenile habitat type. Then, we use single nucleotide polymorphism (SNP) data obtained by Restriction Associated DNA (RAD) sequencing to search for allele frequency changes caused by a single generation of spatially varying selection between habitats. We found evidence for moderate selection operating at multiple loci showing subtle allele frequency shifts between groups of marine and brackish juveniles. We identified subsets of candidate outlier SNPs that, in interaction with habitat type, additively explain up to 3.8% of the variance in juvenile growth and 8.7% in juvenile condition; these SNPs also explained significant fraction of growth rate in an independent larval sample. Our results indicate that selective mortality across environments during early-life stages involves complex trade-offs between alternative growth strategies.

## 1. Introduction

Understanding how species adapt to heterogeneous environments is a central objective in evolutionary biology and ecology [1,2,3,4]. This basic question has important ramifications for our understanding of diversification and extinction and therefore has gained practical importance for predicting species resilience in the face of rapid global change [5,6]. Population genomic approaches facilitate the identification and characterization of adaptive genetic variation in nature [7,8,9,10]. However, deciphering the complex mechanisms underlying local adaptation remains a challenging issue that needs to explicitly consider the links connecting genotype to phenotype and fitness across environments [11].

The main obstacle to establishing such connections occurs when selection affects complex quantitative traits that are themselves encoded by many genes [12,13]. In these situations, genome-wide association (GWA) studies between complex traits and single nucleotide polymorphisms (SNPs) usually lack the power to detect loci with small individual effects on phenotype [14,15]. Moreover, the small allele frequency changes generated by polygenic selection represent a major challenge for ecological genomics studies that search for single locus signatures of selection in molecular data [13,16]. 

Polygenic selection may however generate heterogeneous patterns with both subtle and large allele frequency changes [17,18,19], in particular when local selection occurs in an unpredictable, heterogeneous environment [9]. Indeed, when dispersal occurs on a large scale compared to environmental variation, genotypes are exposed to spatially varying selective pressures, which maintain polygenic variation among individuals [20,21]. Under high gene flow conditions, the migration–selection balance tends to favor intermediate and large effect loci that better resist gene swamping [3,22]. It is therefore likely that, under such conditions, the outlier loci detected in genome scans for selection collectively explain a significant (although incomplete) fraction of the genetic variance for fitness traits [19]. Unfortunately, the joint contribution of candidate variants to phenotypic variation is rarely assessed in empirical genome scan studies, although some studies have proved its interest for connecting genotype to phenotype and fitness [23,24,25]. Here, we implement this approach in a high gene flow marine fish species, to test whether phenotypic differentiation established within a single generation reflects differential survival of genotypes at loci affecting traits, as opposed to pure phenotypic plasticity.

Our model, the gilthead sea bream (*Sparus aurata* L.), is a seasonal migratory fish which uses highly heterogeneous habitats throughout its life cycle. Adults reproduce at sea during winter and the 3–4-month-long larval duration ensures efficient mixing of drifting larvae before they recruit to coastal juvenile habitats [26]. In the Gulf of Lion (northwestern Mediterranean), sea bream postlarvae settle in various types of nursery habitats in March–April without evidence for habitat choice and probably remain within the same habitat over the entire summer before returning to the open sea at the juvenile stage in October–November when water temperatures drop [26,27]. Multielemental otolith fingerprints have revealed that, in the Gulf of Lion, only 15% of the first-year juveniles stay in the nearshore marine habitat, whereas the majority of juveniles use shallow brackish lagoons (47%) or deep marine coastal lagoons (38%) for foraging [27,28]. Marine lagoons have a low food productivity compared to brackish ones but offer much more stable conditions with regards to variations in physicochemical parameters (e.g., oxygen, temperature, salinity) [29]. This trade-off between food availability and environmental disturbance translates into phenotypic differences (including growth rate, condition and shape) between juveniles foraging in different habitats, which rapidly appear a few months after recruitment [26]. The genetic basis of these phenotypic differences and the possible role of diversifying selection remain unclear, although genotype-by-environment interactions have been detected at growth-related candidate genes [30,31].

In this study, we specifically investigated the genotype–phenotype–fitness links across different juvenile environments by combining population genomics and quantitative genetics approaches. To this aim, we jointly analyzed phenotypic and genomic variation in a single cohort of wild sea bream, including newly settled larvae and juveniles that foraged in contrasted (brackish vs. marine) lagoon environments. We developed a test to detect allele frequency shifts caused by a single generation of selection and evaluated the extent to which outlier loci additively contribute to survival probability in each environment. Furthermore, we used polygenic scores to test whether spatially varying selection affecting multiple loci partly explains the observed environment-dependent correlation between growth and condition, and the larval growth rate. Our study illustrates how spatially varying selection acting on complex traits involved in alternative growth strategies can help maintain variation at loci affecting survival in different environments. 

## 2. Materials and Methods

### 2.1. Sampling

Wild sea bream were sampled from two different nursery habitats of a single population in the northern part of the Gulf of Lion, Southern France (Figure 1a). We collected three samples from the same year cohort (2013), consisting of (i) newly settled postlarvae (*N* = 44), (ii) young juveniles (0+) from the Mauguio brackish lagoon (*N* = 106) and (iii) young juveniles (0+) from the Thau marine lagoon (*N* = 106), which represent the main nursery habitats in the region. Postlarvae were collected at the entrance of Thau lagoon in early spring, and juveniles were collected at different points in both lagoons during summer and fall of the same year (Appendix A, see Appendix A for full details). Mauguio and Thau lagoons are separated by less than 30 km, a sufficiently small geographical scale to ensure that all individuals belong to the same panmictic population, since genetic homogeneity has been observed at much larger spatial scales in a previous study [30]. This design enabled us to compare genetic diversity within the postlarval pool (i.e., before postsettlement selective mortality) to genetic diversity within two groups of young juveniles that were grown in ecologically contrasted habitats (i.e., selected in different environments). The two sampled brackish and marine nurseries represent different ends of a trade-off between food availability and environmental disturbance that occur in the coastal waters of the Gulf of Lion [27]. The brackish habitat (Mauguio lagoon) has a high primary production (mean annual Chl *a* concentration of 56.1 µg L^−1^) and provides suitable conditions for rapid growth. However, it also shows important temporal variations of physicochemical parameters (temperature, salinity, dissolved oxygen) due to its shallow depth (<1 m) and inputs from continental waters. By contrast, the Thau lagoon provides a stable marine-like habitat, which is less physiologically stressful but also less nutritive (mean Chl *a* concentration of 1.6 µg L^−1^). Its water characteristics are close to seawater, as illustrated by its marine-like community of macrophyte species [29]. 

### 2.2. Scoring of Phenotypes

We collected two phenotypic traits, condition factor and growth, which are generally considered as good indicators of fitness in fish [32]. Each juvenile was measured (total length in mm from head to the end of caudal fin) and weighted (eviscerated weight in g). The relative condition factor (*K*) was calculated as the ratio of its measured weight to the predicted weight obtained from the log–log linear regression of weight against length based on all juvenile samples [33]. Total lengths were centered to mean zero and normalized to unit variance within each habitat for each sampling date to avoid scaling difficulties due to heterochronous sampling. We thus produced a standardized length index capturing interindividual growth differences while controlling for a possible effect of sampling date on total length. Linear correlations were tested between individual standardized length and condition factor in each habitat. Since we found slopes of opposite signs, we specifically tested the interaction between individual standardized length and habitat type (brackish *vs.* marine) for the condition factor response using an analysis of covariance (ANCOVA) in *R*, which allows fitting different slopes and intercepts in each habitat using *lm*(formula = *Condition* ~ *Habitat* × *Std length*).

The predictive accuracy of the standardized index as a surrogate for individual somatic growth rate was evaluated using otolith age reading in 40 juveniles. Twenty fish from each lagoon were prepared following the protocol of [26] to obtain fish age and calculate somatic growth rate in mm per day. Refined readings were also performed in those 40 juveniles to separately estimate the otolith larval growth rate as the average daily width increment of otolith rings (in μm per day) during the first 60 days of larval life and the juvenile summer otolith growth rate as the average daily width increment during June and July. In addition, 30 of the 44 larvae were submitted to refined otolith reading to estimate individual otolith growth rate during the 60 first days of larval life. 

A picture of the left lateral side of 145 randomly selected juveniles was used to perform morphometric measurements in *tpsDig* 2.17 [34]. Photographs were digitized with 22 anatomically homologous landmarks covering the entire body. A generalized Procrustes analysis was performed using the R package *geomorph* [35], and aligned Procrustes coordinates were used to estimate the mean shape of juveniles in each nursery habitat. We then used the R package *vegan* [36] to estimate the extent to which individual shape was influenced by experimental variables using the model *Shape* ~ *Sampling date* + *Habitat* + *Std length* + *Condition*. The significance of each factor was tested using a redundancy analysis (RDA) marginal effects permutation test (1000 permutations). Finally, we analyzed the influence of growth and condition on shape independently from temporal effects using a partial RDA under the model *Shape* ~ *Std length* + *Condition*, removing the effect of *Sampling date*.

### 2.3. RAD Sequencing, Variant Calling and Individual Genotyping

Genomic DNA was isolated from each of the 256 individuals (larvae and juveniles) using the NucleoSpin Tissue Kit (Macherey-Nagel), standardized to 25 ng µL^−1^ and digested with the restriction enzyme *SbfI*-HF. Eight Restriction Associated DNA (RAD) libraries were constructed by multiplexing 32 uniquely barcoded individuals per library, following a protocol adapted from previous studies [37,38]. Each library was then sequenced on a separate lane of an Illumina HiSeq2000 instrument with 101 bp single-end reads.

Illumina reads were demultiplexed and quality filtered using *process_radtags* in *Stacks* [39,40] and subsequently trimmed to 86 bp (Appendix A). Cleaned individual reads were de novo assembled with *ustacks* using a minimum read depth (−m) of 5 × per individual per allele and allowing at most three mismatches (−M) between two alleles for a same locus. We used the ‘bounded error rate’ model with a maximal error rate of 1% for SNP calling. These parameters were optimized in preliminary runs using different individuals (Appendix A). A catalog of loci was then constructed with *cstacks*, allowing at most three mismatches (–N) between alleles within loci. Each individual was finally matched back to the catalog of loci using *sstacks*, and the program *populations* was used to export genotypes using a minimum call rate of 70% in at least two of the three samples and a minor allele frequency (maf) threshold of 1%. Individual genotypes were exported as biallelic SNPs as well as multiallelic haplotypes defined by reads at the scale of single RAD-tags. 

The two polymorphism datasets (SNPs and haplotypes) were further filtered to include only loci with no missing data in at least 90% of the individuals within each of the three samples (larvae, brackish and marine juveniles). We then excluded markers showing significant deviation from Hardy–Weinberg equilibrium within at least one sample using a *P*-value threshold of 10^−3^ in *PLINK* [41]. This filter mainly aims at removing loci showing strong heterozygote excesses or deficiencies and should not significantly interfere with our capacity to subsequently detect loci influenced by section. Indeed, even strong spatially varying selection can be compatible with HWE proportions [42]. Finally, we used a custom script to detect systematic bias in read counts in favor of a given allele across individuals. For each locus, the allele with the lowest overall read count was identified to compute the ratio of lower to higher allele read depth for each heterozygote. Loci showing significant deviation to the expected ratio of 0.5 (one-sided *t*-test, *P*-value threshold of 0.05) were excluded from the datasets.

### 2.4. Genetic Homogeneity among Samples

The overall genetic structure among all individuals was examined using a principal component analysis (PCA) in the R package *adegenet* [43]. Genetic differentiation between larval and juvenile samples and between brackish and marine juveniles was estimated using pairwise *F*_ST_ for each polymorphism dataset (SNPs and haplotypes).

### 2.5. Test for Single-Generation Selection

In order to detect within-generation allele frequency changes that are unlikely to occur by random chance alone, we developed a statistical test based on a previous method that detects selective changes occurring within a single generation [44]. Our approach takes into account two different sources of allele frequency variance, one due to the finite size of the population, which influences the allele frequency spectrum, and the other due to the finite sample size, which influences the level of uncertainty in measuring the real allele frequencies from population samples. We considered a panmictic common gene pool from which two samples of size $N_{1}$ and $N_{2}$ are drawn within the same generation, either at two different times or in two different environments. For a given allele at a given biallelic locus, the observed allele frequency difference between the two samples ($\Delta p=\left|p_{1}-p_{2}\right|$) was compared with the null distribution of Δp expected from random sampling effects (i.e., due to finite sample size effects). The mathematical details of the method are described as Appendix A. 

The power of the test was evaluated using simulations and compared to Fisher’s exact test, which is classically used to test for genetic differentiation. We considered a finite panmictic population (N=10,000) from which two samples of size $N_{1}\;$ and $N_{2}$ were drawn within the same generation. Selection only occurred in sample 2 with genotypes’ fitness coefficients $\omega_{AA}=1+s$, $\omega_{Aa}=1$ and $\omega_{aa}=1-s$. The value of Δp was calculated after selective mortality, genetic drift and sampling effects, using 100 simulations for each combination of initial allele frequency and selection coefficient value (p,s). Power was measured as the proportion of tests rejecting the null hypothesis of Δp=0 at a 5% significance level for each combination of p and s values (Appendix A).

The test for detecting single-generation selection (SGS) was finally applied to the SNP dataset to compare brackish (*N* = 105) versus marine (*N* = 102) juveniles, using 10,000 iterations to estimate *P*-values. 

### 2.6. Estimating the Survival Probability of Genotypes 

We used the subset of outlier loci detected with the SGS test (*P*-value threshold of 10^−3^) to perform a PCA using all individuals. The distribution of larval genotypes in the plane defined by the first two PC axes was used as a representation of the initial genetic composition within the larval pool before postsettlement selection. This multilocus diversity was then compared to that observed in each juvenile sample in order to estimate the relative enrichment or depletion of genotypes after selection over the genotypic space. For each of the three samples, individual coordinates on the first two PCs were used to perform two-dimensional kernel density estimation using the *kde2D* function in the R package *MASS*. We then calculated the difference between juvenile and larval density estimates to estimate a relative survival probability surface within each habitat. The genotype of each individual was then projected on this surface to get individual survival probability scores in each habitat.

### 2.7. Genotype–Phenotype Links

Genome-wide association (GWA) study was used to identify RAD markers linked with causative variants underlying variation in standardized length and condition. Quantitative-trait association analysis was performed using three different models in *PLINK* [41]: (i) a simple linear model capturing the additive effect of each individual SNP on the phenotype $y=\beta_{0}+\beta_{1}x+\varepsilon$; (ii) a linear model including juvenile environment as a covariate $y=\beta_{0}+\beta_{1}x+\beta_{2}E+\varepsilon$; and (iii) a linear model including a genotype-by-environment interaction term $y=\beta_{0}+\beta_{1}x+\beta_{2\;}E+\beta_{3\;}G\times E+\varepsilon$. We used the *P*-value of the SNP term for the first two models (i.e., with or without adjustment for environmental effects) and the interaction term *P*-value for the third model (i.e., the significance of the difference between the two regression coefficients in each habitat). The genome-wide significance threshold was adjusted using Bonferroni correction by taking within-RAD-tag linkage disequilibrium into account. Quantile–quantile (Q–Q) plots and Manhattan plots were drawn using the R package *qqman*. 

In order to test whether selected genes collectively contribute to phenotypic variation in interaction with the environment, we performed ANCOVA based on individual polygenic scores for both growth using *lm*(formula = *Std length* ~ *Habitat* × *Polygenic Score*) and condition using *lm*(formula = *Condition* ~ *Habitat* × *Polygenic Score*). Individual polygenic scores were obtained by summing over all significant SNPs the number of alleles that are favored in a given environment. We thus obtained a marine polygenic score and a brackish polygenic score that measure the cumulative effects of alleles that were inferred to be advantageous in the marine and brackish habitat, respectively. Because of the low amount of missing genotypes per individual (3.7%), we did not correct for missing data in the calculation of individual polygenic scores. Therefore, marine and brackish polygenic scores (that sum to 2 times the number of loci in the absence of missing data) were computed separately for each individual and separately tested in the ANCOVA. Individual polygenic scores were computed for different sets of loci that were detected at different *P*-value cutoffs in the SGS test to determine the nominal significance threshold maximizing the amount of total phenotypic variance explained by the ANCOVA for each trait [19].

The genotype-by-environment interaction for growth was then tested using the juvenile summer otolith growth rate measured as the average daily width increment (in μm per day) during 60 days in June–July. Although this strongly decreased the power of the tests due to data availability for otolith growth rates in only 40 samples, it provided a better estimation of summer growth as compared to the standardized length. 

Finally, we evaluated the genotype–phenotype relationships using an independent sample of larvae that were not used for outlier SNP detection. Mean otolith growth rate from 30 larvae during the 60 first days of larval life were used to test for association between larval growth and polygenic scores using the linear model *lm*(formula= *Larval growth* ~ *Polygenic Score*). 

## 3. Results

### 3.1. Phenotypic Variation

The average condition factor did not differ significantly between brackish and marine juveniles (marine $\hat{K}$ = 0.996, brackish $\hat{K}$ = 1.011, *t*-test *P* = 0.2). However, the ANCOVA model for the condition response was significant (*P* = 0.014, $R^{2}$= 0.050, Figure 1b) and revealed significantly different regression slopes of individual condition versus standardized length between habitats (*P* = 0.003). Condition tended to be positively correlated with standardized length in the brackish habitat, but negatively correlated in the marine habitat. Otolith daily increments and age readings in 40 juveniles confirmed that the standardized length provides a good substitute measure for both individual somatic growth rate ($R^{2}$= 0.77, Figure 1c) and otolith growth rate calculated from birth to sampling date ($R^{2}$= 0.40). Therefore, standardized length was used as a measure of somatic growth in the remaining analyses, since this phenotype was available for all juvenile fish, which was not the case for otolith readings. For the larval sample, a particular effort was made to measure the daily width increment of otolith rings during the first 60 days of larval life, which averaged to 1.84 ± 0.37 μm per day. This refined measure of larval growth rate was obtained in 30 individuals to test for genotype–phenotype correlations in the larval sample.

Mean body shape differed significantly between habitats (Figure 1d), as revealed by the permutation test for Procrustes distances between groups (*P* < 0.001). On average, brackish juveniles displayed a larger body height between first dorsal spine and anterior pectoral-fin insertion compared with marine juveniles. All factors had significant marginal effects in the RDA of body shape (*P* < 0.001). The effect of standardized length and condition remained significant (*P* < 0.001) after controlling for sampling date. The two axes of the partial RDA constrained by standardized length and condition after removing the sampling date effect explained 7.8% of the total morphological variance, and partly separated brackish and marine juveniles (Figure 1e). In the marine sample, the direction of maximum variance among individuals coincided with RDA axis 2, highlighting a negative correlation between standardized length and condition. In the brackish habitat, individual projections were preferentially distributed along the vector indicating the main gradient of variation in standardized length. 

### 3.2. Population Genetic Homogeneity

A total of 34,679 SNPs (17,579 haplotype markers) were retained after filtering for genotype quality. We detected no evidence of population structure from each of these two datasets (SNPs and haplotypes). Genetic differentiation (*F*_ST_) was almost zero in all pairwise sample comparisons (Appendix A). Genetic homogeneity was also illustrated by the perfect overlap of all three samples in the PCA (Appendix A). 

### 3.3. Genotype–Fitness Links

The test to detect allele frequency shifts caused by single generation selection (SGS) outperformed Fisher’s exact test for genetic differentiation in most simulations (Appendix A). Applying the SGS test between brackish and marine juveniles detected 67 outliers at the nominal significance threshold of 10^−3^ (Figure 2). For most of these SNPs (86.6%), the major allele frequency observed in the larval sample was intermediate to the values observed in juveniles from brackish and marine habitats, and for 71.6% of them at least one juvenile sample lay outside the 90% posterior credibility interval estimated from the larval sample. Also, for the majority of these outliers (59.7%), the major allele increased in frequency in the brackish habitat compared to the larval sample. For simplicity, we later refer to those alleles as “brackish alleles”, and we refer to “marine alleles” in the same manner. 

The PCA based on the 67 outlier loci partly distinguished brackish and marine juveniles along the first axis, whereas larvae occupied mostly intermediate positions (Appendix A). The estimation of relative survival probability scores in each habitat (Figure 3a,b) revealed that 61.4% of the genotypic combinations initially present in larvae were found overrepresented in one of the two habitats at the juvenile stage. This overrepresentation was stronger in the brackish environment, where 45.5% of the genotypes initially present in the larval pool had a positive survival score, whereas only 15.9% were favored in the marine environment. Estimated survival probability scores of larvae in each habitat displayed a moderate trade-off (Figure 3c). The number of brackish alleles per individual (i.e., brackish polygenic score calculated for the 67 outlier loci) was a good predictor of juvenile survival probability scores. Higher brackish polygenic scores were associated with higher survival scores in the brackish habitat but lower survival scores in the marine habitat (Figure 3d; ANCOVA: *P* = 2.8 × 10^−10^, $R^{2}$= 0.208; interaction term: *P* = 6.6 × 10^−11^).

### 3.4. Genotype–Phenotype Links

Only four SNPs showed significant associations at the genome-wide significance level (*P* < 2.88 × 10^−5^) in the GWA study performed for standardized length and condition using three different models. Overlaid Q–Q plots for the three models showed that smaller *P*-values were generally obtained with the model including genotype-by-environment interaction (Appendix A). Among the 669 loci that were detected by the SGS test at a nominal significance thresholds of *P* = 0.01, 15 were found associated with standardized length and 17 with condition using the same *P*-value cutoff.

The proportion of phenotypic variance cumulatively explained by nominally significant variants detected by the SGS test first increased with decreasing nominal significance thresholds and then decreased (Figure 4a,c). The amount of explained variance for standardized length was maximized for a nominal significance threshold of *P* = 0.0075, at which 495 loci cumulatively explained 3.85% of phenotypic variation in interaction with habitat using the marine polygenic score (Figure 4b; ANCOVA: *P* = 0.046; interaction term: *P* = 6.5 × 10^−3^). A similar trend was obtained using measures of mean summer otolith growth rate from 40 juveniles and a nominal significance threshold of *P* = 0.01 to detect candidate selected SNPs. However, the significance of the interaction between the marine polygenic score and habitat was only suggestive (interaction term: *P* = 0.077) due to a lack of power.

The amount of explained variance for condition was maximized for 67 outlier loci (nominal significance threshold of *P* = 0.001), with 8.71% of phenotypic variation explained using the brackish polygenic score in interaction with habitat (Figure 4d; ANCOVA: *P* = 3.4 × 10^−4^; interaction term: *P* = 1.7 × 10^−4^). 

Finally, the independent sample of larvae showed a significant positive correlation between the number of alleles favored in the marine environment at the juvenile stage (calculated using the 11 most significant outliers detected at a nominal significance threshold of *P* = 0.0001) and the larval growth rate calculated during the first 60 days of life (Figure 5, *P* = 0.01, $R^{2}$= 0.214).

## 4. Discussion

How fitness trade-offs across spatially variable environments translate into population phenotypic and genetic responses remains a challenging question in evolutionary biology. Here, we evaluated the single-generation effects of dispersal and local selection in heterogeneous environments by comparing phenotypic and genomic variation among habitats and life stages within a single population cohort of gilthead sea bream. 

Our first observation is that the type of nursery habitat used by juvenile fish influences the direction of the correlation between individual growth rate and condition, which are two important fitness-related traits in fish [32]. These environment-dependent trait correlations are also detectable at the morphometric level, especially in the marine habitat where rapid-growth morphologies tend to display lower condition. Environmental conditions experienced by juvenile fish strongly influence their growth trajectories, which themselves impact a series of fitness-related traits across the entire life cycle (e.g., juvenile mortality, timing of winter migration at sea, survival to sexual maturity, reproduction). Although selection should generally favor higher juvenile growth rates to maximize survival and reproduction, fast growth can be sometimes associated to physiological, developmental and ecological conditions that induce fitness costs [45]. For instance, elevated growth rates can reduce the juveniles' ability to endure periods of starvation, which are typically more frequent in the marine environment. Therefore, the optimal growth strategy may change according to habitat type, and different growth strategies that are genetically encoded may evolve and segregate in the population in response to fitness trade-offs between habitats. 

Although the phenotypic correlations observed here may partly reflect environmentally induced plasticity, they also mirror environment-dependent genetic correlations that have been already observed between growth and condition in experimental conditions [46]. These correlations also correspond to long-term evolutionary responses that have been associated with different life-history strategies in fishes. In general, species that occur in unstable but highly productive environments like brackish lagoons tend to show higher juvenile growth rates and condition indices compared to marine species occupying more stable but less productive environments [47]. Therefore, the sea beam that uses both habitat types at the juvenile stage may face trade-offs between growth and other life-history traits due to varying environmental constraints among nursery habitats. This raises interesting questions about maintenance of variation at loci affecting survival in each environment.

Growth rate and condition are both complex polygenic traits with moderate to high heritable components in sea bream [48,49]. These traits are also genetically correlated [46,49] and are therefore possibly encoded by partially overlapping sets of genes. If selection underlies the environment-dependent trait correlations observed here, this should involve mutations with environment-dependent pleiotropic effects, that is, genotype–environment interactions [50,51]. Here, conditionally neutral mutations and mutations with antagonistic fitness effects across environments [9] were searched by explicitly considering the single-generation footprint of selection in panmixia. As for other statistical approaches based on single-locus tests, our method lacks the power to detect very small allele frequency changes caused by selection. This limitation is however potentially compensated for by the prediction that, in high gene flow species, local selection in spatially heterogeneous environments tends to favor polygenic architectures characterized by a high variance in locus effect size [3,22]. If this prediction is verified, even in the presence of false positives, our approach should at least detect the fraction of loci that contribute the most to differential survival in the fraction of the genome tagged by our RAD markers.

The fitness effects of candidate outlier SNPs were evaluated both individually and collectively. In both cases, comparisons between pre- and postselection samples revealed that groups of juveniles from different environments depart in opposite directions from their larval pool of origin. This suggests that recruitment to brackish and marine nurseries is random with respect to the ability of larvae to succeed in a given environment. Under the alternative hypothesis of habitat selection, our larval sample collected at the entrance of the Thau lagoon would be expected to show more genetic proximity to marine juveniles at outlier loci, which is not what we observed (we actually found a trend in the opposite direction). This clearly rejects the matching habitat choice hypothesis, which was already dismissed in earlier works in sea bream [30]. Therefore, the subtle changes in allele frequency detected here most likely reflect the action of spatially varying polygenic selection spread across multiple loci [13,52]. Interestingly, our results suggest that selection generates softer allele frequency changes in the brackish environment, which is the most abundant habitat type used by the majority of juveniles in the studied region [27,53]. The mean fitness of the population is therefore probably closer to the optimum of the brackish environment, as illustrated by survival probability score surfaces and by the fact that twice as many larvae have a positive survival score in the brackish compared to the marine environment. In species like sea bream, density regulation should mainly occur within nursery habitats, such that each habitat has a constant contribution to the next generation that reflects its carrying capacity [1]. Under weak to moderate trade-offs, this type of life cycle is known to favor the evolution of a single generalist showing intermediate local adaptation biased toward the most abundant and productive habitat [54]. Our results are consistent with this theoretical prediction. A limitation, though, could be the lack of replication compared to other study designs in similar studies (e.g., [25]). Our study only includes a single sample from each of the two alternative juvenile habitats, which is not enough to fully demonstrate that the differences observed are due to the environmental differences between marine and brackish habitats. Therefore, the relationships detected here cannot be extended beyond the two study sites at the present time and will need further investigations.

Our next objective was to evaluate the extent to which candidate SNPs for spatially varying selection explained phenotypic variation among individuals and environments. GWA analyses revealed that individual variants explain at most a small and usually insignificant fraction of phenotypic variance. Therefore, there is a limited overlap between the list of candidate SNPs for spatially varying selection and candidate loci detected in our GWA study. To compensate for this lack of power, we used additive polygenic scores to estimate the joint contribution of candidate variants to phenotypic variation. This approach has already provided informative assessments of polygenic gene action on fitness-related traits [23,24]. It is however intrinsically constrained by the statistical threshold used for candidate SNP detection in single locus tests. GWA studies for complex traits in humans have dealt with this issue using decreasing significance thresholds to progressively include additional smaller-effect candidate variants in polygenic scores [55,56]. Here, we adopted a similar strategy by evaluating the amount of phenotypic variance collectively explained by candidate outlier SNPs that were detected without taking phenotypic information into account, under different significant thresholds. Although this approach may be prone to false-positive detection of candidate SNPs for selection, false positive outliers are unlikely to contribute to phenotypic variance by chance, thus insuring an independent assessment of the additive effect of outlier SNPs on phenotype in interaction with environment. Moreover, the correlation detected between the marine polygenic score and larval growth rate indicated that false positive detection was not a major issue. Indeed, outlier loci were only searched using juvenile samples, and therefore the larval sample provided a completely independent support that at least some of the detected genes are truly involved in the studied phenotypes. These results also indicate that larval growth and juvenile growth are controlled by partially overlapping sets of genes, which is not surprising. 

Overall, our results provide empirical support that selected variants with moderate to small individual effects on fitness traits can cumulatively explain several percent of the phenotypic variance among individuals living in contrasted environments. More generally, they support the view that selected mutations with antagonistic pleiotropic effects partly underlie the environment-dependent correlations between growth and condition in juvenile sea bream.

Antagonistic pleiotropy between environments likely involves complex allocation trade-offs that emerge due to compromises between growth and stress regulation [57,58]. Juvenile sea bream are subject to different sources of stress, ranging from highly unstable conditions in brackish lagoons to frequent starvation in the marine environment [26]. This supposes the existence of different optimal growth trajectories among environments, because the fast-growing genotypes favored in rich habitats are not the best adapted to poor habitats, where they are unable to cope with prolonged food deprivation. The well-studied phenomenon of compensatory growth [59] illustrates the necessity for fish to perform growth regulation to buffer unpredictable environmental variation. In juvenile sea bream, the environment-dependent correlation between growth and condition may thus reflect selection for different growth trajectories in relation with food availability and stress. 

Reduced genome representation methods such as RAD-Seq have been criticized as providing insufficiently dense genome-scans for detecting local adaptation genes when linkage disequilibrium extends over small chromosomal distances around selected loci [60,61,62]. We acknowledge that we have most probably missed a number of selected loci, possibly some with larger effects than those we detected. This can be explained by several reasons, including imperfect linkage disequilibrium of marker loci with the causative variants, or the existence of rare variants of large effect and especially of many small effect-size loci that cannot be detected with the limited sample sizes used in this study. This drawback adds to the issue discussed above that even intermediate-effect variants are expected to display small allele frequency differentials between habitats. However, our approach allowed quantifying the proportion of phenotypic variance explained by the candidate outlier loci [19], which proved to be non-negligible. In line with previous studies [63,64], we therefore argue that the cost-effective approach implemented here provides sufficient marker density to get access to a meaningful fraction of the genetic variance for fitness traits.

## 5. Conclusions

To conclude, the molecular footprint of local polygenic selection acting within a single generation was found at multiple SNPs, which will need to be further validated using a recently developed high-density SNP array in the gilthead sea bream. Our results imply that fitness-related traits such as juvenile growth and condition are encoded by multiple small-effect genes with antagonistic pleiotropic effects across environments. They also support the view that different life-history strategies, which are partly genetically encoded, segregate in sea bream as a response to different optimal growth trajectories across juvenile habitats.

## Figures and Tables

**Figure 1 genes-11-00398-f001:**
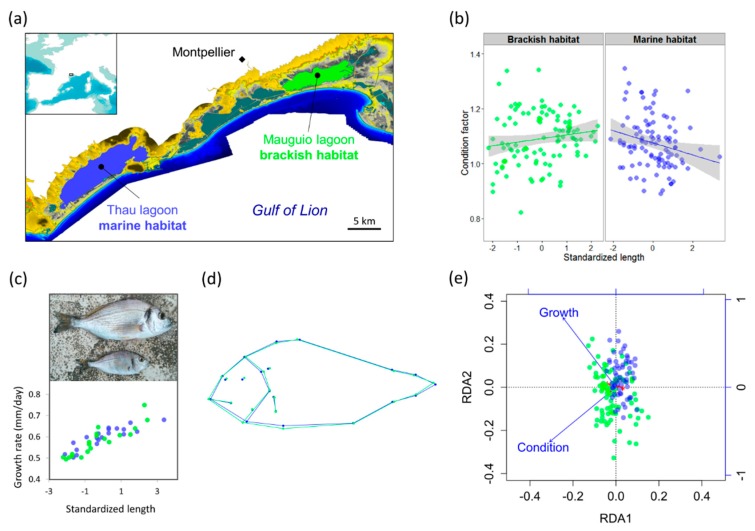
Phenotypic variation of juvenile sea bream from different nursery habitats. (**a**) Juveniles of the same year cohort were sampled in two closely located nursery habitats in the Gulf of Lion: A stable but low-productivity marine habitat (Thau lagoon, blue) and a stressful but highly productive brackish habitat (Mauguio lagoon, green). (**b**) Habitat-dependent correlation between individual standardized length and condition factor (ANCOVA interaction term: *P* = 0.003, shaded areas indicate 95% confidence intervals). (**c**) Predictive accuracy of standardized length for somatic growth rate estimated through otolith age reading in 20 individuals from each habitat ($R^{2}\;$
= 0.77). (**d**) Mean body shape differences between habitats (permutation test: *P* < 0.001). (**e**) Partial redundancy analysis (RDA) of body shape controlling for sampling date, showing the main directions of morphological variation related to growth and condition.

**Figure 2 genes-11-00398-f002:**
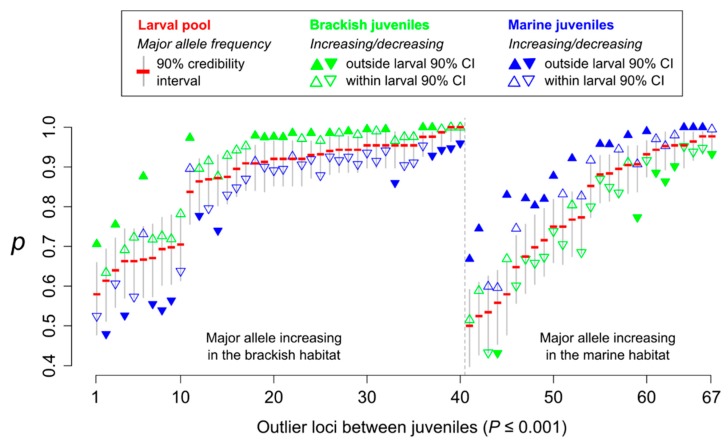
Allele frequency changes at 67 outlier loci detected between brackish and marine juveniles with the single generation selection (SGS) test, using a nominal significance threshold of 10^−3^. For each locus, allele frequency of the major allele (*p*) is indicated in the larval pool (red, with the 90% credibility interval in grey), brackish juveniles (green triangles) and marine juveniles (blue triangles). Upper triangles show increased allele frequency compared to the larval pool; lower triangles show decreased allele frequency compared to the larval pool.

**Figure 3 genes-11-00398-f003:**
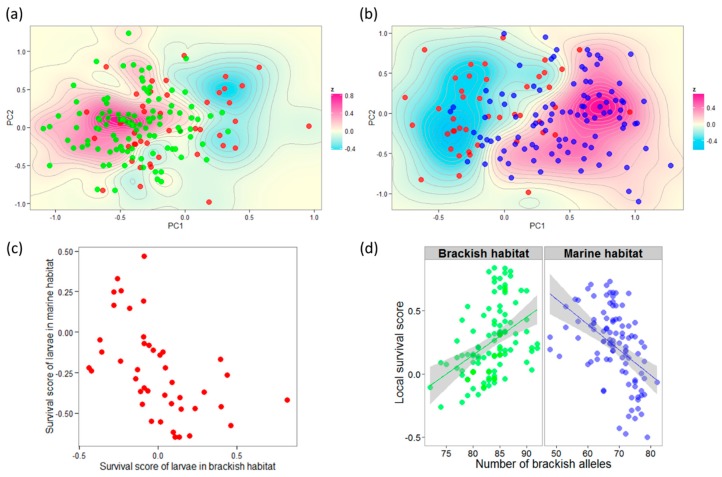
Genotype–fitness relationships assessed with 67 outlier loci detected with the SGS test. Individual coordinates in the two first PCA axes were compared (**a**) between larvae (red) and brackish juveniles (green) or (**b**) between larvae and marine juveniles (blue) to estimate the relative enrichment or depletion of multilocus genotypes over the genotypic space (*z*, the relative survival probability score surface) in each habitat. (**c**) Estimated survival scores of larvae show a moderate trade-off between habitats. (**d**) The sum of brackish-favored alleles in individual juveniles is positively correlated with survival score in the brackish habitat but negatively correlated in the marine habitat (ANCOVA interaction term: *P* = 6.6 × 10^−11^).

**Figure 4 genes-11-00398-f004:**
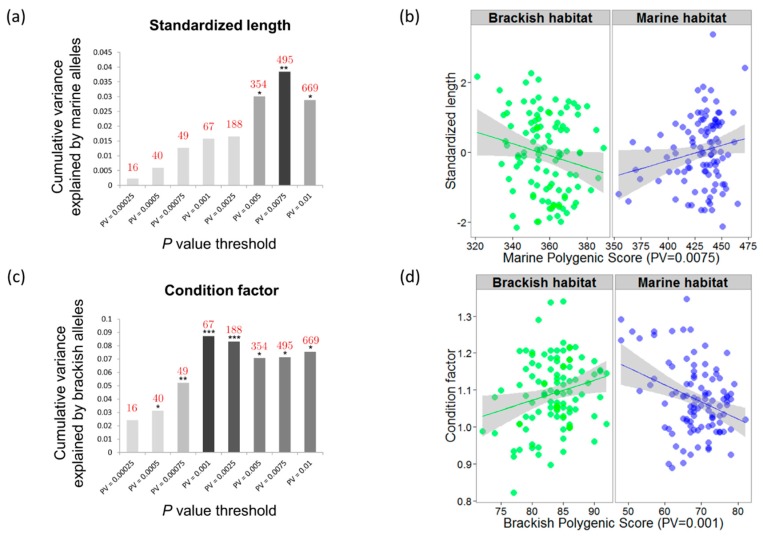
Phenotypic variance explained by outlier loci. (**a**) The cumulative variance in standardized length (y axis) explained by marine-favored alleles increases when outlier single nucleotide polymorphisms (SNPs) reaching lower significance thresholds (x axis) are included in the ANCOVA model (interaction term: *P* < 0.05 *, < 0.01 **, < 0.001 ***). The number of outlier SNPs corresponding to each significance threshold appears in red. The maximum amount of explained phenotypic variance is attained for a total of 495 outliers detected at a nominal significance threshold of 0.0075. (**b**) Genotype–environment interaction for standardized length assessed with an ANCOVA model using individual marine polygenic scores calculated with 495 outliers ($R^{2}\;$
= 0.0385, interaction term: *P* = 6.5 × 10^−3^). (**c**) The cumulative variance in condition explained by brackish-favored alleles is maximized for 67 outlier loci detected at a *P*-value threshold of 0.001. (**d**) Genotype–environment interaction for condition assessed with an ANCOVA model using individual brackish polygenic scores calculated with 67 outliers ($R^{2}\;$= 0.0871, interaction term: *P* = 1.7 × 10^−4^).

**Figure 5 genes-11-00398-f005:**
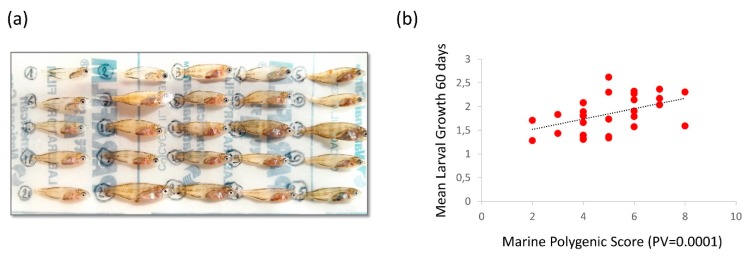
Phenotypic variance explained by outlier loci in the independent sample of larvae. (**a**) Picture of larvae collected on the same day, showing differences in size between individuals. (**b**) Positive correlation ($R^{2}\;$
= 0.214, slope *P* = 0.01) between the average daily width increment of otolith rings during the first 60 days of larval life (y axis, in μm per day) and the marine polygenic score calculated for the 11 most significant outliers found in the SGS test (x axis, *P*-value threshold of 0.0001).

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
