# Peer review of "Within-Generation Polygenic Selection Shapes Fitness-Related Traits across Environments in Juvenile Sea Bream"

_genes, 2020, doi:10.3390/genes11040398_

Round 1
Reviewer 1 Report
Comments for Authors:
In this article Rey and colleagues studied Juvenile Sea Bream (Sparus aurata) from two different habitats from Gulf of Lion. Two different traits likely related to fitness in the two habitats are analyzed in a single generation. Authors could identify that the two studied traits show opposite correlations in the two habitat and multiple RAD loci are associated with fitness. The overall study is interesting and could shed light on polygenic adaptations, but it is very complex and technical, without any explicit attempt to read a broad readership beyond Sea Bream or fish community. Specific comments below:
1. There are many interesting elements in the study, but the writing is very complex and technical, which is a major limitation in my view. It is difficult to mention every instance, but every analysis and result is described in a very technical way. Authors should simplify the manuscript keeping the broader evolutionary community in mind and simplify the results.
2. The methods and statistics sound thorough, but there is no rational behind most of the technics used. For example: Why are standardized length and condition factor used? Why are they better then other possible metrics? How are these relevant for adaptations in the two habitats? Similarly, what is the rational to use linear regression? Can other types of regression perform better/worse? Why use otolith age reading to assess predictive accuracy of standardized index? These are only a few mentions. Authors should add rational and reasons for the choice of analytical techniques used throughout the methods section. Keep them simple and easy to understand and relate them to the physiology of fish in the habitat.
3. The implications of the results for physiology of fish and adaptations are completely missing. For example, what are the implications of opposite relation between the parameters for the physiology of fish and adaptations at the two habitat? Are the loci that are associated with the studied traits relevant to the adaptations? If yes, how? Again, this should be done to all the results, either when they are described or in the discussion section.
4. It will be necessary to go into more details of the GWAS results. Did authors map the significant SNPs to genes, and observed any genes relevant for growth or local adaptations? This will be a crucial analysis to add to the manuscript.
5. Discussion is extremely technical. Authors should consider to simplify it to cater to broad evolutionary readership in mind, discussing how these results are relevant for GWAS and rapid adaptations.
6. The variance explained by the significant SNPs is low. Authors should discuss what could be other elements that drive adaptations in different habitat. Unobserved SNPs, epigenetic factors etc.
7. Minor points: Fig1c, what does inset picture mean? Are these from two habitat?
Typo at line 400.
Author Response
"Please see the attachment."

Reviewer 2 Report
The paper entitled “Within-Generation Polygenic Selection Shapes Fitness-Related Traits across Environments in Juvenile Sea Bream” analyses phenotypic and genotypic pattern in three sea bream populations (Post-larvae close to marine Thau lagoon, and juveniles at marine and brackish habitat).
The paper give us information about phenotypical traits (length, that is standardized, condition, shape and otolith width related to somatic growth). It reveals different regression slopes of individual condition versus standardized length between habitats. Therefore, authors establish this equation Condition ~ Habitat * Std length. One of my concern is that they are comparing different ages in brackish and marine populations, because in marine population almost half of the fish are much younger. And maybe also the slope can change depending on the age. Maybe otolith growth can be used to estimate the age. Authors should justify this point. The results related to otolith and growth are explained very quickly.
Regarding genetic information, authors demonstrate genetic Homogeneity among Samples through genetic differentiation (FST) and by the perfect overlap of all three samples in PCA. They describes the test for detecting single-generation selection (SGS) and detect 67 outliers between brackish and marine juveniles detected, that they use to estimate the survival probability of genotypes. Polygenic Score was calculated by summing over all significant SNPs and authors demonstrate that loci cumulatively explained phenotypic variation.
From my point of view, paper contains a lot of information, sometimes it is explained very quickly and sometimes it is written in a particular way, for example the aim of the paper is written as a result (line 87 – 95). In the end, this work offer us a valid approach of the genetic variance for fitness traits, in addition to phenotypical results.
However, I would like to know how these different ages in marine juvenile population can affect to these results. Since authors compare juveniles with larvae population to detect outliers, in marine population there is half animals closer to post-larvae stage than juvenile stage, although these animals are in the lagoon at that time. Maybe it is not important for genetic comparison but for phenotypical result could be, thus this should be clarified.
Therefore, my main concern is to justify how the different ages in marine population can affect the results.
Specific comments
M&M
131 How length is measured from the head to the fork or to the caudal fin? It should detailed in this section.
132-133 Condition factor, it is not clear how is it measured. Authors cite “log-log linear regression of weight against length based on all the individuals sampled (Le Cren 1951)”, but are they changing the slope depending on the sample? If not, which slope they use? Wx100/L3???
Line 137 “Since we found slopes of opposite signs” This is part of the results.
Line 145-149 It is confusing. How many larvae and juveniles were analysed for refined readings? “In addition, 30 larvae were submitted to otolith reading to estimate individual otolith growth rate during the 60 first days of larval life” Are the same larvae than the previous one?
Results
Line 280 Instead of “size” is more accurate “length”
Discussion
Line 405 – 406 There is more current works. M. García-Celdrán, G. Ramis, M. Manchado, A. Estévez, J.M. Afonso, E. Armero 2015 Estimates of heritabilities and genetic correlations of carcass quality traits in a reared gilthead sea bream (Sparus aurata L.) population sourced from three broodstocks along the Spanish coasts. Aquaculture
Author Response
R.2.1. One of my concern is that they are comparing different ages in brackish and marine populations, because in marine population almost half of the fish are much younger. And maybe also the slope can change depending on the age. Maybe otolith growth can be used to estimate the age. Authors should justify this point. The results related to otolith and growth are explained very quickly.
>> The effect of differences in sampling dates on juvenile length was accounted for by standardizing length values within each sample collected at a given date. We added the following sentence to the text: “We thus produced a standardized length index capturing inter-individual growth differences while controlling for a possible effect of sampling date on total length.” (L121-124).
We agree that growth rate calculated from otolith age reading would make a more precise estimation of somatic growth, but the reading of daily rings increments is extremely demanding and expansive. Therefore, we only performed such reading in 40 individuals, from which we verified that the standardized length was a reasonably good proxy for the growth rate calculated from otolith age reading (Fig. 1C, R²=0.77). This has been justified L267-271.
Additional details concerning otolith growth rates and their use in subsequent analyses were included in the revision (L269-275).
R.2.2. From my point of view, paper contains a lot of information, sometimes it is explained very quickly and sometimes it is written in a particular way, for example the aim of the paper is written as a result (line 87 – 95).
>> The beginning of the last paragraph of the introduction has been rephrased in a less direct style (L77-79).
R.2.3. In the end, this work offer us a valid approach of the genetic variance for fitness traits, in addition to phenotypical results. However, I would like to know how these different ages in marine juvenile population can affect to these results. Since authors compare juveniles with larvae population to detect outliers, in marine population there is half animals closer to post-larvae stage than juvenile stage, although these animals are in the lagoon at that time. Maybe it is not important for genetic comparison but for phenotypical result could be, thus this should be clarified. Therefore, my main concern is to justify how the different ages in marine population can affect the results.
>> We are not sure to fully understand this criticism, since the detection of outlier loci was only applied to juvenile samples (L205-207): “The test for detecting single-generation selection (SGS) was finally applied to the SNP dataset to compare brackish (N = 105) versus marine (N = 102) juveniles, using 10,000 iterations to estimate P-values.”
Allele frequencies in the larval sample are shown in Figure 2 in comparison to juvenile samples just to show that, as expected, larval frequencies lie in between the allele frequencies of juveniles in the two habitats.
The studied population is panmictic, so the use of two different life stages within a same cohort (generation) mostly aims at illustrating how selection drives allele frequencies apart from the frequencies in the larval pool between the larval and juvenile stage. In any case, this does not affect the finding of genotype-phenotype correlations in juveniles (Fig. 4), nor in the larval sample (Fig. 5).
Specific comments
M&M
131 How length is measured from the head to the fork or to the caudal fin? It should detailed in this section.
>> We now mention “from head to the end of caudal fin” (L117-118)
132-133 Condition factor, it is not clear how is it measured. Authors cite “log-log linear regression of weight against length based on all the individuals sampled (Le Cren 1951)”, but are they changing the slope depending on the sample? If not, which slope they use? Wx100/L3???
>> We now specify that the slope is based on the log-log linear regression based on all juvenile samples (L120)
Line 137 “Since we found slopes of opposite signs” This is part of the results.
>> This explanation is necessary to understand the methodology, so we kept this sentence in the M&M section
Line 145-149 It is confusing. How many larvae and juveniles were analysed for refined readings? “In addition, 30 larvae were submitted to otolith reading to estimate individual otolith growth rate during the 60 first days of larval life” Are the same larvae than the previous one?
>> We now specified that it concerns the 40 juveniles analyzed with otolith readings (L132), and 30 of the 44 larvae (L135)
Results
Line 280 Instead of “size” is more accurate “length”
>> Done (L266-267)
Discussion
Line 405 – 406 There is more current works. M. García-Celdrán, G. Ramis, M. Manchado, A. Estévez, J.M. Afonso, E. Armero 2015 Estimates of heritabilities and genetic correlations of carcass quality traits in a reared gilthead sea bream (Sparus aurata L.) population sourced from three broodstocks along the Spanish coasts. Aquaculture
>> True, but the genetic correlation between condition factor and length is not significant in this study, which is why this reference has not been included.
We thank you reviewer #2 for your helpful comments.

Reviewer 3 Report
I certainly read the manuscript entitled “Within-Generation Polygenic Selection Shapes Fitness-Related Traits across Environments in Juvenile Sea Bream”. This research is challenging to find genotype-environment interaction in a panmixia species, gilthead sea bream. The content of article seemed fine, but the weakest point of this research is using single year samples from two alternative juvenile habitats, as you mentioned in the discussion section.
Please consider my comments below to improve the manuscript.
>2.3.
Have you done Stacks analysis using reference-based mapping or de novo? In the result section, I found you mapped RAD-tags onto the reference genome of sea bream, but no information was present in the M&M section. Please add a bit detail of in silico analysis.
SNPs were pruned using HWE threshold (P-value threshold of 10-3) at least one sample. How about SNP filtering of HWE only in the larval population and used remained loci, which is no HWE in the larval population, for further genomic analysis. This may improve the oultlier finding.
>2.5.
Haven’t you done outlier analysis using common software such as LOSITAN and BayesScan? If you have done, please show the result in supplementary data.
>3.2.
You showed the pairwise FST, which clearly supported the PCA. But, I think you also need to show the global FST and AMOVA to be sure.
>3.3.
I didn’t find any information of 67 outlier loci, such as LG number, position, and possible candidate gene. Could you provide this kind information in supplementary data?
Author Response
"Please see the attachment."
